# Skin Lesion Classification Using Hybrid Convolutional Neural Network with Edge, Color, and Texture Information

Changmin Kim [1], Myeongsoo Jang [2], Younghwan Han [2], Yousik Hong [2] and Woobeom Lee [2,*]

1 AI Software Education Institute, Soonchunhyang University, Asan 31538, Republic of Korea; kcmin212@nate.com
2 Department of Information Communication Software Engineering, Sangji University, Wonju 26339, Republic of Korea; sksaudtn@nate.com (M.J.); yhhan@sangji.ac.kr (Younghwan Han)
* Correspondence: beomlee@sangji.ac.kr

**Abstract:** Herein, a new paradigm based on deep learning was proposed that allows the extraction of fine-grained differences between skin lesions in pixel units for high accuracy classification of skin lesions. As basic feature information for a dermoscopic image of a skin region, 50 different features were extracted based on the edge, color, and texture features of the skin lesion image. For the edge features, a line-segment-type analysis algorithm was used, wherein the visual information of a dermoscopic image was precisely analyzed in terms of the units of pixels and was transformed into a structured pattern. Regarding the color features of skin lesions, the dermoscopic image was transformed into multiple color models, and the features were acquired by analyzing histograms showing information regarding the distribution of pixel intensities. Subsequently, texture features were extracted by applying the well-known Law's texture energy measure algorithm. Feature data ($50 \times 256$) generated via the feature extraction process above were used to classify skin lesions via a one-dimensional (1D) convolution layer-based classification model. Because the architecture of the designed model comprises parallel 1D convolution layers, fine-grained features of the dermoscopic image can be identified using different parameters. To evaluate the performance of the proposed method, datasets from the 2017 and 2018 International Skin Imaging Collaboration were used. A comparison of results yielded by well-known classification models and other models reported in the literature show the superiority of the proposed model. Additionally, the proposed method achieves an accuracy exceeding 88%.

**Keywords:** skin lesion classification; hybrid convolutional neural network; CNN; feature detection; ISIC dataset

## 1. Introduction

Medical AI for aiding the diagnosis and prediction of diseases has been actively investigated to provide useful information for experts and increase diagnostic accuracy [1–3]. Many types of medical imaging techniques exist, such as X-ray, computed tomography, and magnetic resonance imaging. Images obtained from medical imaging exhibit high resolution and are suitable for analysis requiring high accuracy and precision. High-resolution images allow minimal feature loss during image preprocessing, such as segmentation and expSansion. In the case of skin lesion images, high-resolution images are effective for diagnosis and provide useful information. In addition, for enhanced visibility, dermatologists use high-quality magnifying dermoscopic lenses and a powerful light sources for in-vivo observations of the skin surface [4,5]. Skin diseases exhibit similar features to other types of diseases; thus, an observation method as effective as using human eyes is required. This paper presents an observation method and a classification model for the classification of seven types of lesions (actinic keratosis, basal cell carcinoma, benign keratosis-like lesions, dermatofibroma, malignant melanoma, melanocytic nevi, vascular lesions) in various skin conditions.

Skin diseases are typically diagnosed during health checkups, where a biopsy is obtained for a more detailed analysis. During a health checkup, a dermatologist visually observes the skin surface using a high-quality dermoscope and performs a diagnosis of the condition via medical consultation. If an abnormal finding is identified during the health checkup, then some or all of the lesions are removed to perform a more precise and detailed examination in a laboratory. The overall process is extremely time-consuming and costly, and the diagnosis result is fault-prone as it depends on the subjective bias of the dermatologist [6]. To address the limitations of human inspection, deep learning-based classification system utilizing big data has been proposed, e.g., ImageNet [7], MSCOCO [8], and PASCAL [9], which demonstrate high performance in terms of the detection and diagnosis of skin lesions [10–13].

During the visual observation of an object through human eyes, a significant amount of information is transmitted to the brain to allow a rapid recognition of the object [14]. In this process, numerous features of the object, such as its edge, color, and texture, are analyzed; subsequently, the information is described. Furthermore, by conducting an observation closely, a more detailed analysis can be conducted on the features of the object, allowing one to identify and understand the essence of the object. The key to accomplishing such meticulous observation is to obtain numerous features from a single image using a range of feature detectors, followed by combining the extracted features to derive the desired result. Gonzale-Luna et al. [15] extracted 137 features of breast lesions based morphological categories of shape, orientation, margin, echo pattern, and posterior. Subsequently, these extracted features were applied to machine learning algorithms, i.e., AdaBoost, K-nearest neighbors (kNNs), linear discriminant analysis, multilayer perceptron (MLP), radial basis function network, random forest, and support vector machine (SVM), and the results showed a classification performance. Derdour et al. [16] proposed a system for recognizing handwritten digits using a combination of different invariant feature extraction methods and multiple classifiers; in particular, cavities, Zernike moments, Hu moments, and the histogram of gradient were used for feature extraction. This study aimed to improve the accuracy of recognizing handwritten digits in the modified National Institute of Standards and Technology database using a Tree, a kNN, an MLP, an SVMOVO, and an SVMOVA. López-Iñesta et al. [17] proposed a hybrid approach that combines both feature extraction and feature expansion to improve classification performance based on similarity and metric learning. Feature extraction and expansion were performed using training samples obtained from data transformation and a set of standard distances. The method was used for preprocessing input data into an SVM, and the performance of the proposed method was demonstrated via four different evaluation experiments using face and object recognition data.

Based on previous studies, meticulous observations performed via machine learning typically involves the use of two or more feature detectors to extract different feature information. However, these previous studies primarily adopted hybrid approaches or systems with combined or integrated feature detector models. In this case, the throughput of the detectors and machine learning models must be high; as such, sophisticated equipment and high energy are required. The features derived from the models may be difficult to analyze depending on the detector used, and the hyperparameters of all detectors must be analyzed, which is often challenging. In general, since hyperparameters affect the final results and performance, they can only be tuned optimally by performing a series of experiments and time-consuming process.

In this paper, we focused on the basic components of images through relatively simple and easy feature analysis instead of using multiple detectors or performing large-scale models and complex processing as in previous studies [7–17]. These components are essential features used in the diagnosis of skin lesions, and we aimed to extract them to reduce the scale and complexity of neural models. In addition, the size of skin lesion images can always be reduced to small, fixed-size data for clear representation of the processing. Shape features based on edge information allow the analysis of the visual appearance of

skin lesions, color features allow the observation of skin color changes due to disease, and texture features allow the detection of surface abnormalities of the skin caused by disease. To extract these various types of feature information in this study, shape, color, and texture features were extracted using an edge detector. Subsequently, color space conversion was performed and Law's texture energy measure (LTEM) texture detection filter [18] was used to augment the features. Consequently, 16 edge features were generated to extract the shape information of skin lesions, and nine feature vectors were generated for color features through color space (RGB, HSV, and YCrCb) [19–21] conversion and histogram analysis for each channel of the respective color space. In addition, by obtaining 25 different texture features using the LTEM filter, 50 visual features were generated to be used for model training. In this case, to reduce the computational load required when a single dermoscopic image was augmented into 50 features, the augmented 50 visual features were converted into structured data via line-segment-type analysis [22,23] and histogram analysis. The converted data were used as inputs of a deep learning model for skin lesion classification, and experiments were conducted on feature augmentation and classification models.

This paper introduces existing studies on skin lesion classification in Section 2, describes feature extraction and diagnosis process in Section 3, presents the performance evaluation of the proposed method in Section 4, and finally concludes the paper with Section 5.

## 2. Related Studies

Skin lesions show similar appearances to a congenital nevus (mole), which poses difficulties in the early diagnosis of skin diseases via visual inspection by humans. Moreover, differences among various types of skin diseases and lesions are difficult to differentiate, which necessitates a precise analysis of skin lesions [24].

(a)  Actinic keratosis (AKIEC) presents as scaling pink/red macules or papules on areas of chronic sun exposure and is a precancerous condition of the skin that may evolve into squamous cell carcinoma (SCC).

(b)  Basal cell carcinoma (BCC), a malignant neoplasm derived from non-keratinizing cells originating in the epidermis and its basal layer, is locally invasive, and its metastasis is rare. BCC and SCC are the two most common subtypes of nonmelanoma skin cancer.

(c)  Benign keratosis-like lesions (BKLs) are benign tumors that develop on the skin, characterized by excessive formation of keratinized tissue resulting in a thick, irregular epidermal layer. Although they are generally not life-threatening, accurate diagnosis is necessary as they can be mistaken for malignant tumors.

(d)  Dermatofibroma refers to a benign tumor of the skin composed of fibrocytes and fibers forming connective tissue, often occurring in conjunction with other types of skin tumors.

(e)  Malignant melanoma (MEL) is a type of skin tumor caused by the malignant transformation of melanocytes; it may be congenital or acquired. Although MEL primarily develops on the skin, it can originate from melanocytes located in the eyes or in mucosal membranes lining the anus, nose, and esophagus.

(f)  Melanocytic nevi (NV) refer to pigmented lesions of the skin and typically occur in children or adolescents, and occasionally at birth. NV comprise a cluster of melanin-producing cells known as melanocytes. As time progresses, the number of nevi may increase, or those already present may increase in size or become darker.

(g)  Vascular lesions (VASC) are lesions such as nevus flammeus, facial flushing, telangiectasia, and hemangioma, which are caused by hemoglobin oxidation and triggers severe oxidative stress in blood cells and vessels. They appear in various forms, such as facial flushing, nevus flammeus, and rosacea.

If melanocytic nevi is suspected of being malignant melanoma, excisional biopsy is essential and, in the case of BCC or SCC, biopsy is performed to plan subsequent treatment. The examination requires approximately one week and incurs a high cost. To resolve

this problem, researchers are attempting to replace conventional methods of skin disease diagnosis with medical AI [25–27]. Figure 1 shows eight types of skin lesions.

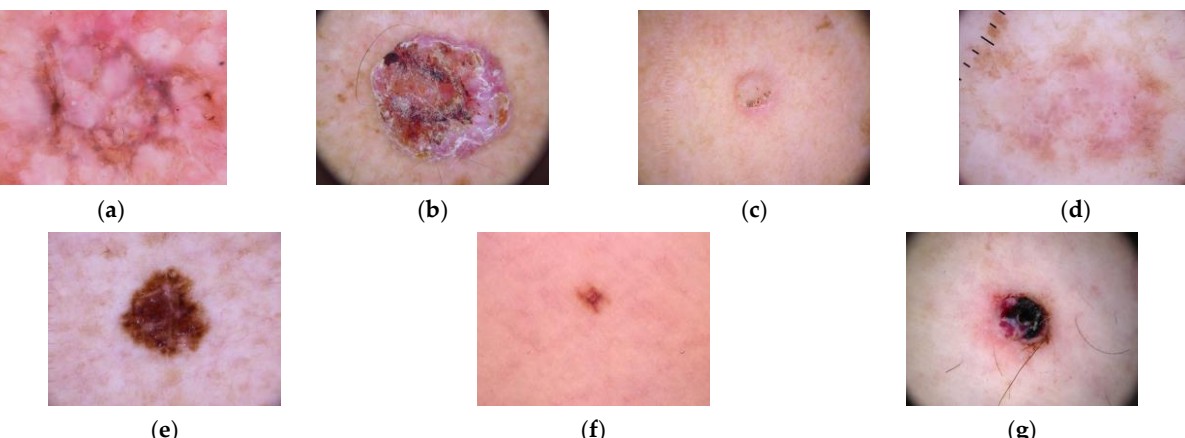

**Figure 1.** Example of skin lesions images; (**a**) actinic keratosis, (**b**) basal cell carcinoma, (**c**) benign keratosis-like, (**d**) dermatofibroma, (**e**) malignant melanoma, (**f**) melanocytic nevi, (**g**) vascular lesions.

AI-Masni et al. [28] proposed an integrated diagnostic framework that combines the skin lesion boundary segmentation stage and the multiple skin lesion classification stage. They used a full resolution convolutional network to segment skin lesion boundaries from whole dermoscopic images; additionally, they used Inception-v3, ResNet-50, Inception-ResNet-v2, and DenseNet-201 to classify segmented skin lesions. The classification performance of each classifier model in terms of accuracy was 88.05%, 89.28%, 87.74%, and 88.70%, respectively, and the final results were obtained by synthesizing the accuracy of these classifiers. Since this framework uses multiple classifiers, the initial setup for experiments incurs high costs and the training and inference processes are time-consuming.

Barata et al. [29] classified skin lesions hierarchically into melanocytic and non-melanocytic lesions based on their origins; subsequently, they were further classified into benign or malignant lesions depending on the malignancy degree. The classification method used may be similar to that used for binary tree problems, and DenseNet-161 and ResNet-Inception were used as the architectures of the input image encoder. For images that were subjected to feature extraction through the image encoder, a long short-term memory (LSTM) network was used to perform the process of the image decoder to derive a hierarchical diagnosis. The final accuracy was 78.0%, and the classification accuracy between melanocytic and nonmelanocytic lesions was 92.2%. Similarly, two models were used in the present study as the image encoder, and an RNN-based LSTM model was used as the decoder. LSTM requires a longer training time and inference time than CNN-based models, owing to its recurrent architecture.

By contrast, the skin disease diagnosis method proposed herein uses only a single classification model, which is achieved by extracting multiple types of features from a single image. By reducing the computational load required for visual feature extraction through dimensionality reduction, computation can be performed effectively using simple equipment. Many types of features require the segmentation of the edge image, color spaces (RGB, HSV, and YCrCb), and texture features, and each information must be converted into new one-dimensional (1D) vector information through line-segment-type analysis and histograms. In this study, 50 features in 1D vector format were generated and, after conversion, a two-dimensional (2D) matrix (50 × 256) was generated. The generated 2D matrix was used as an input for a 1D convolution-based feature analysis model. The methods for detecting each type of feature are relatively simple, and a diagnosis of a skin disease was performed by combining existing well-established methods and the line-segment-type analysis algorithm proposed by the authors in a previous study [22,23]. The proposed processing method allows the problem-solving framework of skin disease

diagnosis to evolve from a classification model-centered diagnostic method (used in the past) to a feature analysis-centered diagnostic method.

## 3. Feature Extraction from Dermoscopic Images

In this study, to extract edge, color, and texture features from a dermoscopic image for skin lesion classification, fine-grained edge detection algorithm, color space conversion, and texture detection using the LTEM filter were performed.

### 3.1. Extraction of Edge Feature Information via Line-Segment-Type Analysis

To perform high-precision analysis of visual characteristics in skin lesions, we used edge detection methods that can extract features from small internal areas, rather than the typical contour processing methods.

Figure 2 shows the borders of a skin lesion. The outer-border (OB) represents the border between the normal skin region and the lesion, and the inner-border (IB) inside the lesion represents the border information in the center of the lesion. Since conventional edge detectors identify edge features for strong line segments regardless of the location of the skin lesion in the outer or inner borders, small areas in the IB tend to be removed, which results in reduced accuracy in feature analysis. Therefore, in this study, when detecting the edge information of a skin lesion, the brightness was removed to differentiate the OB and IB of the lesion. In general, lesions show a weaker intensity (darker) than normal skin regions, and a threshold value for removing areas other than the lesion can be calculated using Equation (1).

$$
\begin{aligned}
q1 &= \text{MIN}(I) + \left( \sqrt{\frac{\sum I(x,y)^2}{N \times M} - \left( \frac{\sum I(x,y)}{N \times M} \right)^2} \right) \\
q2 &= \text{MAX}(I) - \left( \sqrt{\frac{\sum I(x,y)^2}{N \times M} - \left( \frac{\sum I(x,y)}{N \times M} \right)^2} \right)
\end{aligned}
\tag{1}
$$

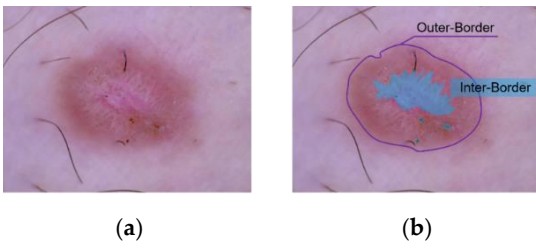

| (a) | (b) |
| --- | --- |

**Figure 2.** Examples of border features; (**a**) original image, (**b**) border feature.

$I(x,y)$ in Equation (1) denotes the intensity of the image at position $(x,y)$ of the dermoscopic image, and $N \times M$ indicates total number of pixels, which represents the size of the image. After calculating the minimum and maximum values of the $I(x,y)$ pixel of the input dermoscopic image, the lower and upper threshold values ($q1$ *and* $q2$, respectively) are generated using the standard deviation of $I(x,y)$. The values of $q1$ *and* $q2$ are set to be similar to the mean pixel value by adding the standard deviation to the minimum value and subtracting the standard deviation from the maximum value.

Figure 3a presents the skin lesion image and Figure 3b presents the histogram of Figure 3a. In Figure 3a, the normal skin appears brighter than the lesion and belongs to the uppermost section of the histogram, whereas the darker region in the lesion belongs to the lowest section of the histogram. Using the threshold range $[q1, q2]$, the uppermost and lowermost regions of the input image can be removed using Equation (2).

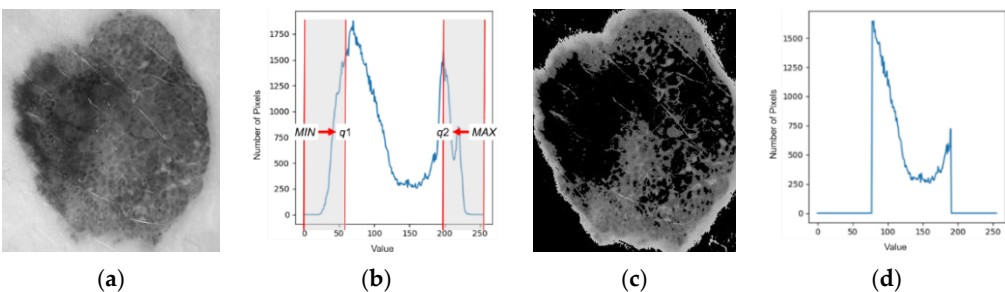

**Figure 3.** Example of a histogram for skin lesion image; (**a**) original image, (**b**) histogram of original image, (**c**) thresholding image, (**d**) histogram of thresholding image.

Based on Equation (2), if the pixel value of $I(x, y)$ is within the threshold range $[q1, q2]$, then the original brightness value at the corresponding position is assigned; if the pixel value is out of the range, then it is set to 0, and the threshold image $I_T(\cdot)$ is generated. Consequently, the OB and IB of the lesion are clearly revealed as the normal skin region, and the regions showing low brightness inside the lesion are removed, as shown in Figure 3c. Figure 3d presents the histogram of $I_T(\cdot)$, which is an image with the threshold range values applied. As shown, the uppermost and lowermost regions have been removed in the histogram.

$$I_T(x, y) = \begin{cases} I(x, y) & \text{if } I[x, y] \text{ exsits in } [q1, q2] \\ 0 & \text{otherwise} \end{cases}. \tag{2}$$

Otsu's algorithm [30] was applied to $I_T(\cdot)$ in Figure 3c, where some regions of brightness were removed via thresholding, and the IB and OB were extracted. Otsu's algorithm is a method for dividing an object region in an image. Using a histogram, the algorithm allows the intensity to be calculated to achieve the most optimal classification of the image into two classes.

Algorithm 1 is a method of detecting IB and OB regions based on Otsu's algorithm. The optimal intensity $k$ is calculated using Otsu's algorithm. Subsequently, $\alpha$ is calculated using the mean $m_G$ and standard deviation $\sqrt{\sigma_G^2}$ of the image $I_T(\cdot)$, and the optimal $k$ and $\bar{k}$ are generated. Figure 4a shows the skin lesion image, and the image obtained after applying image thresholding using Equations (1) and (2) is shown in Figure 4b. When the $k$ derived using Otsu's algorithm is applied to image (b), the OB information is obtained as shown in Figure 4c. Subsequently, by applying $\bar{k}$, the IB image is obtained, as shown in Figure 4d. Figure 4e,f show edge images detected by the Canny edge detector from the images shown in Figure 4c,d, respectively. Figure 4e shows the extraction of the IB region, and Figure 4f shows the line-segment feature of the OB region. Finally, the image shown in Figure 4g is generated by the XOR operation of the images shown in Figure 4e,f.

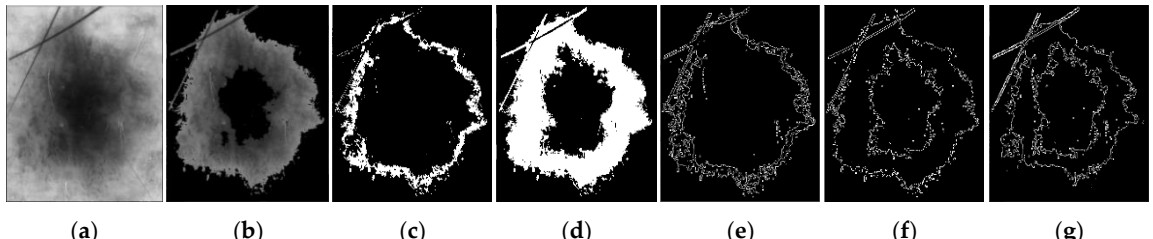

**Figure 4.** Edge feature collected through inter & outer-border extraction and canny edge detector; (**a**) skin lesion image, i($\cdot$), (**b**) thresholding image by $[q_1, q_2]$, $I_T(\cdot)$, (**c**) Ostu $- k$ image, $I_B^O(\cdot)$, (**d**) Ostu $- \bar{k}$ image, $I_B^I(\cdot)$, (**e**) Ostu $- k$ edge image, $I_E^O(\cdot)$, (**f**) Ostu $- \bar{k}$ edge image, $I_E^O(\cdot)$, (**g**) XOR image $I_E(\cdot)$ of (**e**,**f**).

In Figure 4b, the normal skin region was removed via the previous processing; however, the normal skin region remained around the lesion. Therefore, Otsu's algorithm was used to distinguish between the normal skin region and lesion such that only the normal skin region is extracted, as shown in Figure 4c. As for the regional separation, the boundary lines closer to the center represent the characteristics inside the lesion, while the boundary lines further away from the center form the outer shape of the lesion. After extracting the edges from the images shown in Figure 4c,d using the Canny detector (i.e., (c) -> (e), (d) -> (f)), an XOR operation was performed to remove the border of the normal skin region, which is the common area in the two images, and the resulting image is presented in Figure 4g.

As shown in Figure 4g, when an edge image with visual shape information of the IB and OB for the skin lesion was generated, line-segment-type analysis was performed in pixel units using a detector filter based on Equation (3).

---

**Algorithm 1:** Inner and Outer Border Division Algorithm Using Otsu's Algorithm

Input: $I_T(x,y)$ Image with removed brightness via thresholding $I(x,y)$; image size, $N \times M$
Output: Inner and outer border-image $I_B(x,y)$ and edge image $I_E(x,y)$ of $I_B(x,y)$

① Generate an array $hist[L]$ of size $L$ to create a histogram distribution diagram
② Use the brightness intensity of input $I_T(x,y)$ as the index of array $hist[L]$ and increase the value of the corresponding index by 1.
③ Repeat Step ② to scan all the pixels of the input $I_T(x,y)$.
④ For array $hist[i]$ that has completed step ③, obtain the normalized histogram value is using $p_i = \frac{hist[i]}{N \times M}$.
⑤ Calculate the parameters required for the classification into two classes:

- The probability that a specific pixel belongs to Class 1 $P_1(k) = \sum_{i=0}^{k} p_i$
- The probability that a specific pixel belongs to Class 2 $P_2(k) = 1 - P_1(k)$
- The mean of the brightness of pixels in Class 1, $m_1(k) = \frac{1}{P_1(k)} \sum_{i=0}^{k} ip_i$
- The mean of the brightness of pixels in Class 2, $m_2(k) = \frac{1}{P_2(k)} \sum_{i=k+1}^{L-1} ip_i$,
- The mean intensity up to the intensity level k, $m(k) = \sum_{i=0}^{k} ip_i$
- The global mean of the image, $I_T(x,y)$ $m_G = \sum_{i=0}^{L-1} ip_i$
- The global variance of the image, $I_T(x,y)$ $\sigma_G^2(k) = \sum_{i=0}^{L-1} (i - m_G)^2 p_i$

⑥ Using the between-class variance $\sigma_B^2(k) = \frac{[m_G P_1(k) - m(k)]^2}{P_1(k)[1 - P_1(k)]}$, obtain the value of $k$ for maximizing the variance.

⑦ Using the optimal value of k calculated in Step ⑥, derive $\bar{k} = \frac{k}{\alpha}$ using $\alpha = \frac{m_G}{\sqrt{\sigma_G^2}}$.

⑧ $I_B^O(x,y)$ is the image of the outer-border division through k in $I_T(x,y)$, $I_B^I(x,y)$ is the image of inner-border division through $\bar{k}$

⑨ Create edge image $I_E^O(x,y)$ of $I_B^O(x,y)$ and edge image $I_E^I(x,y)$ of $I_B^I(x,y)$

⑩ Perform an XOR operation on the two edge images to generate the inner- and outer-border edge image $I_E(x,y)$.

---

$$LTDF = \begin{bmatrix} 1 & 2 & 4 \\ 8 & 0 & 16 \\ 32 & 64 & 128 \end{bmatrix}. \tag{3}$$

The line-segment-type detector filter (LTDF) expressed in Equation (3) is a detector filter, the coefficients of which are composed of values of $2^N$. Furthermore, a convolution operation with the binary edge image generated for edge information extraction was performed with pixel 1 in the image. As an example, Figure 5a shows a partial region in the input image.

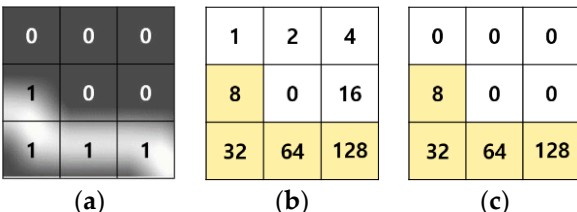

**Figure 5.** Classification of line-segment types using a *LTDF*; (**a**) Some data from input Figure 4g, (**b**) responsive coefficient of *LTDF*, (**c**) responsive coefficient.

When convolution is performed with the LTDF shown in Figure 5b, the shaded coefficient values in Figure 5b are used in the operation. In other words, the visual line-segment information in the image can be classified into a binary code format in a structured pattern (e.g., [8, 32, 64, 128]), as shown in Figure 5c, and the filtering result can be used as a unique type number for the corresponding line segment.

In this study, the visual information converted to structured information using the LTDF was classified into 256 types, as shown in Table 1. When the structured information was classified into similar line-segment types, the classification result can be organized as presented in Table 2.

**Table 1.** Line-segment type no. corresponding to the response of LTDF; EN: Eigen No., N-A: Non-activity, P: Point, H: Horizontality, V: Verticality, C: Curve, D: Diagonal, F: Face, P-n: Pattern-n.

| No. | Feature | No. | Feature | No. | Feature | No. | Feature | No. | Feature | No. | Feature | No. | Feature | No. | Feature |
|---|---|---|---|---|---|---|---|---|---|---|---|---|---|---|---|
| 0 | N-A | 32 | P | 64 | P | 96 | H | 128 | P | 160 | P-1 | 192 | H | 224 | H |
| 1 | P | 33 | P-1 | 65 | P-4 | 97 | P-2 | 129 | P-1 | 161 | C | 193 | P-5 | 225 | P-6 |
| 2 | P | 34 | P-4 | 66 | P-1 | 98 | P-2 | 130 | P-4 | 162 | P-7 | 194 | P-2 | 226 | P-12 |
| 3 | H | 35 | P-2 | 67 | P-2 | 99 | P-3 | 131 | P-5 | 163 | P-9 | 195 | P-13 | 227 | P-14 |
| 4 | P | 36 | P-1 | 68 | P-4 | 100 | P-5 | 132 | P-1 | 164 | C | 196 | P-2 | 228 | P-6 |
| 5 | P-1 | 37 | C | 69 | P-7 | 101 | P-9 | 133 | C | 165 | P-15 | 197 | P-9 | 229 | P-19 |
| 6 | H | 38 | P-5 | 70 | P-2 | 102 | P-13 | 134 | P-2 | 166 | P-9 | 198 | P-3 | 230 | P-14 |
| 7 | H | 39 | P-6 | 71 | P-12 | 103 | P-14 | 135 | P-6 | 167 | P-19 | 199 | P-14 | 231 | P-18 |
| 8 | P | 40 | V | 72 | D | 104 | C | 136 | P-4 | 168 | P-2 | 200 | C | 232 | C |
| 9 | V | 41 | V | 73 | C | 105 | C | 137 | P-5 | 169 | P-6 | 201 | C | 233 | C |
| 10 | D | 42 | C | 74 | C | 106 | C | 138 | P-7 | 170 | P-8 | 202 | P-9 | 234 | C |
| 11 | C | 43 | C | 75 | C | 107 | C | 139 | P-10 | 171 | P-11 | 203 | C | 235 | C |
| 12 | P-4 | 44 | P-5 | 76 | P-2 | 108 | P-10 | 140 | P-7 | 172 | P-9 | 204 | P-8 | 236 | P-11 |
| 13 | P-2 | 45 | P-6 | 77 | P-8 | 109 | P-11 | 141 | P-9 | 173 | P-19 | 205 | P-17 | 237 | P-16 |
| 14 | C | 46 | C | 78 | P-9 | 110 | C | 142 | P-8 | 174 | P-17 | 206 | C | 238 | C |
| 15 | C | 47 | C | 79 | C | 111 | C | 143 | P-11 | 175 | P-16 | 207 | C | 239 | C |
| 16 | P | 48 | P-4 | 80 | D | 112 | C | 144 | V | 176 | P-2 | 208 | C | 240 | C |
| 17 | P-4 | 49 | P-7 | 81 | P-7 | 113 | P-8 | 145 | P-5 | 177 | P-9 | 209 | P-10 | 241 | P-11 |
| 18 | D | 50 | P-2 | 82 | C | 114 | P-9 | 146 | C | 178 | P-8 | 210 | C | 242 | C |
| 19 | C | 51 | P-8 | 83 | P-9 | 115 | C | 147 | C | 179 | P-17 | 211 | C | 243 | C |
| 20 | V | 52 | P-5 | 84 | C | 116 | C | 148 | V | 180 | P-6 | 212 | C | 244 | C |
| 21 | C | 53 | P-9 | 85 | P-8 | 117 | P-17 | 149 | P-6 | 181 | P-19 | 213 | P-11 | 245 | P-16 |
| 22 | C | 54 | P-10 | 86 | C | 118 | C | 150 | C | 182 | P-11 | 214 | C | 246 | C |
| 23 | C | 55 | P-11 | 87 | C | 119 | C | 151 | C | 183 | P-16 | 215 | C | 247 | C |
| 24 | P-1 | 56 | P-2 | 88 | C | 120 | C | 152 | P-2 | 184 | P-3 | 216 | C | 248 | C |
| 25 | P-2 | 57 | P-12 | 89 | P-9 | 121 | C | 153 | P-13 | 185 | P-14 | 217 | C | 249 | C |
| 26 | C | 58 | P-9 | 90 | P-15 | 122 | C | 154 | P-9 | 186 | C | 218 | C | 250 | P-16 |

**Table 1.** *Cont.*

| No. | Feature | No. | Feature | No. | Feature | No. | Feature | No. | Feature | No. | Feature | No. | Feature | No. | Feature |
|-----|---------|-----|---------|-----|---------|-----|---------|-----|---------|-----|---------|-----|---------|-----|---------|
| 27 | C | 59 | C | 91 | C | 123 | P-16 | 155 | C | 187 | C | 219 | P-18 | 251 | C |
| 28 | P-2 | 60 | P-13 | 92 | P-9 | 124 | C | 156 | P-12 | 188 | P-14 | 220 | C | 252 | C |
| 29 | P-3 | 61 | P-14 | 93 | C | 125 | C | 157 | P-14 | 189 | P-18 | 221 | C | 253 | C |
| 30 | C | 62 | C | 94 | C | 126 | P-18 | 158 | C | 190 | C | 222 | P-16 | 254 | C |
| 31 | C | 63 | C | 95 | P-16 | 127 | C | 159 | C | 191 | C | 223 | C | 255 | F |

**Table 2.** Visual line-segment type summary about Engen type in Table 1; TN: type no.

| TN | Type | No. of Type | TN | Type | No. of Type |
|----|------|-------------|----|------|-------------|
| 1 | Non-Activity (N-A) | 1 | 14 | Pattern7 (P-7) | 8 |
| 2 | Point (P) | 8 | 15 | Pattern8 (P-8) | 4 |
| 3 | Verticality (V) | 6 | 16 | Pattern9 (P-9) | 8 |
| 4 | Horizontality (H) | 6 | 17 | Pattern10 (P-10) | 2 |
| 5 | Diagonal (D) | 4 | 18 | Pattern11 (P-11) | 17 |
| 6 | Curve (C) | 97 | 19 | Pattern12 (P-12) | 8 |
| 7 | Face (F) | 1 | 20 | Pattern13 (P-13) | 4 |
| 8 | Pattern1 (P-1) | 6 | 21 | Pattern14 (P-14) | 4 |
| 9 | Pattern2 (P-2) | 8 | 22 | Pattern15 (P-15) | 4 |
| 10 | Pattern3 (P-3) | 16 | 23 | Pattern16 (P-16) | 4 |
| 11 | Pattern4 (P-4) | 4 | 24 | Pattern17 (P-17) | 8 |
| 12 | Pattern5 (P-5) | 8 | 25 | Pattern18 (P-18) | 8 |
| 13 | Pattern6 (P-6) | 4 | 26 | Pattern19 (P-19) | 8 |

Table 1 shows the line-segment types that can be classified using the LTDF, and the numbers in Table 1 indicate the unique numbers (eigen-numbers) based on the line-segment type. For the same line-segment type, different eigen-numbers are assigned depending on the position. Meanwhile, general line-segment types, such as vertical, horizontal, and curved, as well as unusual line types such as checkered, are expressed as Pattern *N*. A total of 19 different patterns were generated, as shown in Figure 6.

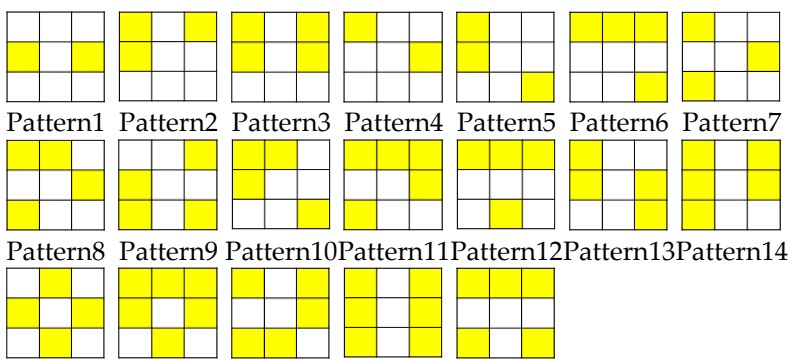

**Figure 6.** Pattern-n feature of the response coefficient through the LTDF.

In this study, the values calculated using the LTDF and filtering were set as the eigen-numbers to reduce the visual edge information in the image. At this time, the input dermoscopic image was segmented into regions of the same area based on Equation (4), and the LTDF was applied to each segmented region. The pixel value in the segmented image, which is the value obtained using the LTDF, corresponds to the eigen-number in Table 1.

In Equation (4), both *M and N* denote the size of the input image $I_E(\cdot)$, $\alpha$ indicates the number of vertical and horizontal segmented regions of equal area, and $\lfloor \cdot \rfloor$ denotes the floor() function. In this study, a single image is partitioned into 16 regions, and their

line-segment type value is calculated using the LTDF. For each line-segment type, Equation (5) is used to generate an $EFA_{i,j}[\cdot]$ vector, which is the edge feature for a single segmented region $S_{i,j}(\cdot)$.

$$S_{i,j}(x,y) = I_E[S_w \times i + x, S_h \times j + y]$$
$$\text{where, } S_w = \lfloor M/\alpha \rfloor, S_h = \lfloor N/\alpha \rfloor, 0 \leq x < S_w, 0 \leq y < S_h \tag{4}$$

$$EFA_{i,j}[\cdot] = [a_0, a_1, \ldots, a_k, \ldots, a_K], \ i,j = 0, 1, \ldots, \alpha - 1, K = 255$$
$$\text{where, } a_k = Cum(k), \ k = S_{i,j}(x,y) \circledast LTDF. \tag{5}$$

$EFA_{i,j}[\cdot]$ in Equation (5) represents the reduced visual edge information, *K* the maximum number of line-segment types, and $\circledast$ the filtering operation in the image processing. The function $Cum(\cdot)$ indicates a cumulative function, which returns the cumulative number of pixels whose filtering result obtained using the LTDF is k in the segmented region $S_{i,j}(\cdot)$. After calculating the values of $EFA_{i,j}[\cdot]$ for all the segmented regions of the dermoscopic image, the results were converted into $i \times j \times K$ vectors through normalization and used as edge information for the learning model.

### 3.2. Extraction of Color Feature Information from Color Space Histogram

To extract the color information, which is another visual information of skin lesions, the image was converted to RGB, YCbCr, and HSV color models, which are representative color spaces generally used in object segmentation and classification. Additionally, histogram analysis was performed on each channel of the respective converted color models for feature extraction. The YCbCr color model represents chromaticity components in terms of luminance (Y), blue-difference (Cb), and red-difference (Cr). The HSV model comprises hue (H), saturation (S), and value (V) components, which clearly distinguishes luminance and chrominance, and is typically used in color image segmentation because its color space components are the most intuitive to human recognition. Figure 7 shows an example of a dermoscopic image converted to the three color models.

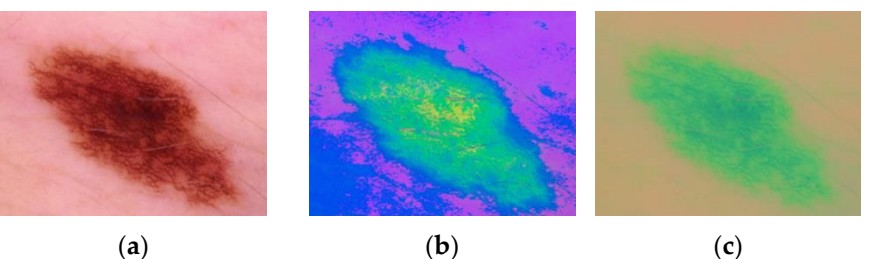

<div align="center">(<b>a</b>)        (<b>b</b>)        (<b>c</b>)</div>

**Figure 7.** Image example of 3-color space; (**a**) RGB image, (**b**) YCrCb image, (**c**) HSV image.

When a dermoscopic image was converted to three color spaces, the chrominance of the lesion was clearly generated in contrast to normal, as shown in Figure 7, and the color information vector $CFM[\cdot]$ was generated as shown in Equation (6) using histogram information.

$$CFA[\cdot] = [C_1(H_R, H_G, H_B), C_1(H_Y, H_{Cr}, H_{Cb}), C_1(H_H, H_S, H_V),]$$

$$\text{where, } H_m = [h_0, \cdots, h_k, \cdots, h_K],$$
$$m = R, G, B, Y, Cr, Cb, H, S, V, (K = 255) \tag{6}$$
$$h_k = Histogram(k).$$

In Equation (6), $CFA[\cdot]$ represents the color information corresponding to three channels for each of the three color models, and it is represented as a vector with a size of 3 (color model) $\times$ 3 (number of channels) $\times$ 256 (histogram), thus resulting in nine vectors with 256 elements. $h_k$ is an element of $H_m[\cdot]$, which is a histogram information vector of one channel and indicates the value of level k in the channel's histogram.

### 3.3. Extraction of Texture Features Using LTEM Filter

The LTEM algorithm is typically used to extract the text features of various objects and allows texture information from dermoscopic images to be extracted with relative simplicity. As shown in Table 3, for masks used to extract texture features from the image presented herein, five-value vectors that can extract texture information such as level, edge, spot, wave, and ripple were used to obtain the masks (as shown in Equation (7)).

$$
M_{i,j} = V_i^T \times V_j = \begin{bmatrix} a_1 \\ a_2 \\ \vdots \\ a_k \end{bmatrix} \times \begin{bmatrix} b_1 & b_2 & \cdots & b_k \end{bmatrix} = \begin{bmatrix} a_1 b_1 & \cdots & a_1 b_k \\ \vdots & \ddots & \vdots \\ a_k b_1 & \cdots & a_k b_k \end{bmatrix} \tag{7}
$$
$$
V_i = [a_1, a_2, \ldots, a_k], V_j = [b_1, b_2, \ldots, b_k].
$$

**Table 3.** Types of LTEM's default masks.

| Label | Texture | 5-Value |
|:---:|:---:|:---:|
| 1 | Level | $V_1(\text{L5}) = [1, 4, 6, 4, 1]$ |
| 2 | Edge | $V_2(\text{E5}) = [-1, -2, 0, 2, 1]$ |
| 3 | Spot | $V_3(\text{S5}) = [-1, 0, 2, 0, -1]$ |
| 4 | Wave | $V_4(\text{W5}) = [-1, 2, 0, -2, 1]$ |
| 5 | Ripple | $V_5(\text{R5}) = [1, -4, 6, -4, 1]$ |

In Equation (7), $V_i[\cdot]$ denotes a mask vector corresponding to label i shown in Table 3, and a 2D spatial filter for texture feature extraction was generated by the vector product of five 1D mask vectors. Here, $M_{i,j}$ denotes a texture feature extraction filter generated by two 1D masks corresponding to labels *i* and *j*. Therefore, the texture feature TFA[·] of the dermoscopic image extracted for learning is defined as shown in Equation (8) below.

$$
TFA[\cdot] = \begin{bmatrix} H_{1,1} & \cdots & H_{1,J} \\ \vdots & \ddots & \vdots \\ H_{I,1} & \cdots & H_{I,J} \end{bmatrix}
$$
$$
\text{where, } if\ i\ and\ j\ is\ 1, \quad H_{i,j} = [h_0, \cdots, h_k, \cdots, h_K],\ K = 255 \tag{8}
$$
$$
h_k = Histogram(k)\ of\ T_{i,j}(x, y) = I_T(x, y) \circledast M_{i,j}
$$
$$
otherwise\ H_{i,j} = EFA[\cdot]\ of\ T_{i,j}(x, y) = I_T(x, y) \circledast M_{i,j}.
$$

In Equation (8), $H_{i,j}[\cdot]$ denotes the value corresponding to the histogram or line-segment-type features of image $T_{i,j}(\cdot)$, which is obtained by extracting texture information from the dermoscopic image by filtering the $I_T(\cdot)$ image through $M_{i,j}$. The value of $M_{1,1}(L5L5)$ is a value that represents relatively high brightness. A feature vector was generated using the histogram value, whereas for the other texture features, the feature vectors were generated using the line-segment-type analysis method, which was discussed in the previous section. Figure 8 shows a filtering result image for 25 texture masks (5 × 5) used for texture feature extraction in this study.

By performing the process above, a feature vector for a skin lesion was generated with sixteen edge features ($EFA[\cdot]$), nine color features ($CFA[\cdot]$), and twenty-five texture features ($TFA[\cdot]$) from a single dermoscopic image, as shown in Figure 9. The generated single feature data exhibit the same vector size with 256 elements. Therefore, the input pattern data of the artificial neural network for skin lesion recognition, $X[\cdot]$, was generated as a single vector of size 50 × 256 and flattened based on Equation (9) below. Subsequently, they were used as the input of the neural network model.

$$X[\cdot] = \begin{bmatrix} EFA \\ CFA \\ TFA \end{bmatrix} = \begin{bmatrix} EFA_{1,1} & \cdots & EFA_{1,4} \\ \vdots & \ddots & \vdots \\ EFA_{4,1} & \cdots & EFA_{4,4} \\ H_R & H_G & H_B \\ H_Y & H_{Cb} & H_{Cr} \\ H_H & H_S & H_V \\ H_{1,1} & \cdots & H_{1,5} \\ \vdots & \ddots & \vdots \\ H_{5,1} & \cdots & H_{5,5} \end{bmatrix} = \begin{bmatrix} EFA_{1,1} & \cdots & EFA_{4,4} H_R & \cdots & H_V H_{1,1} & \cdots & H_{5,5} \end{bmatrix} \qquad (9)$$

**Figure 8.** Examples of texture extraction image $T_{i,j}(\cdot)$ using LTEM mask $M_{i,j}$.

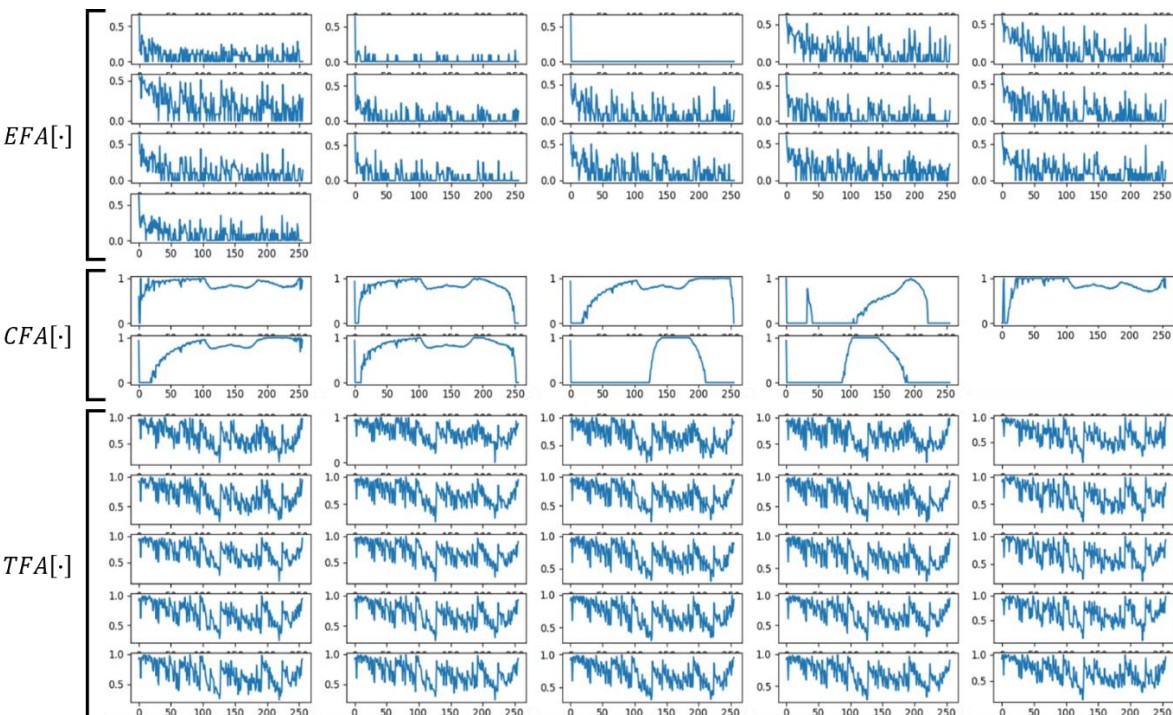

**Figure 9.** Example of extracted EFA[·], CFA[·], and TFA[·] for skin lesion classification.

## 4. Evaluation of Model Performance

In this study, datasets from the 2017/2018 International Skin Imaging Collaboration (ISIC) were used to evaluate the performance of the proposed feature extraction method and learning model used in this study. For the proposed model, which uses structured data generated via feature extraction from a dermoscopic image as training data, the performance of the optimization function was first evaluated based on the learning rate. Subsequently, the model performance was evaluated via comparison with state-of-the-art image classification models and results pertaining to skin lesion classification reported in the literature.

### 4.1. Experimental Setup

As shown in Figure 9, the proposed algorithm extracts edge, color, and texture features from a single dermoscopic image, and then generates data with reduced dimensionalities smaller than 256 × 256, which is the general the data size of a raw image (as shown in Equation (9)). In the generated data, strong feature components shown in the image can be represented with emphasis, and weak feature components can be represented in terms of the number of cumulative pixel intensities, which allows a representation with an aggregated feature component.

In this study, to perform machine learning on the input feature data extracted from a dermoscopic image, as shown in Figure 10a, we established four convolution groups for the model: Conv-Ag, Conv-Bg, Conv-Cg, and Conv-Dg. As shown in Figure 10b, each convolution group was a modular form with a 1D convolution layer connected to a 1D global max-pooling layer, where N modules were not connected to each other but configured in parallel. In this case, each 1D convolution layer had a different kernel size and stride, and the ReLU function was used as an activation function. Among the convolution groups, the padding of A and C was set as "same," and the padding of B and D was set as "valid". For convolution groups A and B, the analysis was performed based on the features of the input data, as shown in Figure 9, whereas for convolution groups C and D, the rows and columns of the input data were transposed to analyze the differences of each feature.

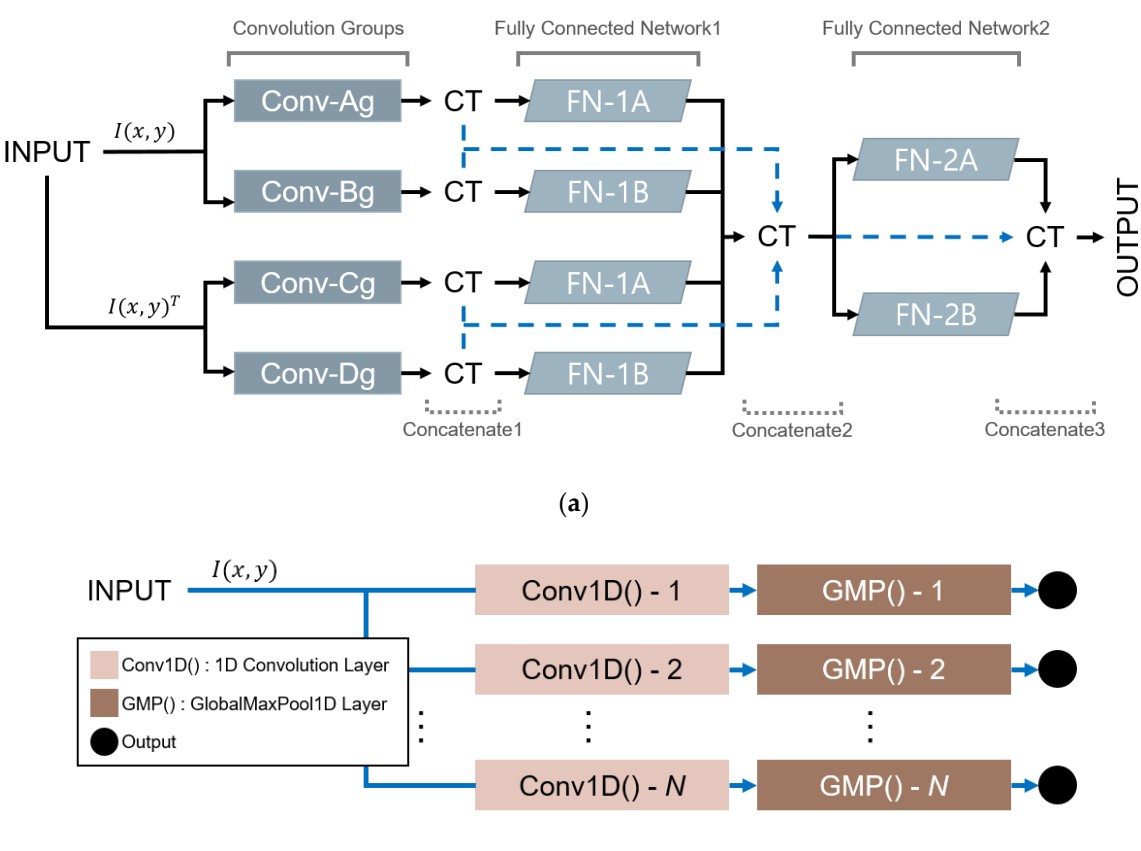

**Figure 10.** A convolutional neural network model for learning a skin lesion. (**a**) Overview of a learning model. (**b**) A convolution group structure.

Each set of data calculated from the convolution groups was concatenated and integrated into one set of data, and after performing learning for each group using fully connected network-1 (FN-1), all data were integrated via a second concatenation. In FN-2, the strong features of the second concatenated data were analyzed. Finally, the data integrated via the third concatenation were classified into the "softmax" type.

In this study, to validate and evaluate the performance of the proposed feature extraction method and learning model, datasets from ISIC 2017 [31] and 2018 [32] were used. In particular, the performance was evaluated based on the annotation information of each set of data obtained from the ISIC datasets. The ISIC 2017 dataset comprised 2000 skin lesion images from three categories: "nevus (1372)", "melanoma (374)", and "seborrheic keratosis (254)." Meanwhile, the ISIC 2018 dataset comprised 10,015 images from seven categories: "actinic keratosis (327)", "basal cell carcinoma (514)", "benign keratosis-like lesions (1099)", "dermatofibroma (115)", "melanoma (1113)", "melanocytic nevi (6705)", and "vascular lesions (142)."

As for the hardware used in the experimental setup, an Intel Core i7-6700 CPU with 32 GB of RAM and GeForce RTX 2080 Ti were used on a Windows 10 operating system. To implement the feature extraction and machine learning algorithms, Numpy1.18 and OpenCV4.1 were used based on Python3.6, Keras2.1, and Tensorflow2.4 libraries.

*4.2. Experimental Results*

Table 4 shows the experimental results for the optimal learning of the proposed model using the ISIC 2018 dataset. Among 10,015 images, 80% were used as the training dataset (8012 images) and 20% as the test dataset (2003 images). The result was measured using the EarlyStopping() function and the Keras-tuner library. Meanwhile, the performance was evaluated at learning rates of 0.01, 0.001, 0.0001, and 0.00001. The result shows that the

AdaGrad function demonstrated the highest accuracy of 0.88 at learning rates of 0.01 and 0.001, and that the loss of the AdaGrad function was insignificant (lr = 0.00001). Figure 11 presents a confusion matrix of the AdaGrad optimization function (lr = 0.001).

**Table 4.** Accuracy and loss of optimization ft. in proposed model on the ISIC 2018 datasets.

| Optimization ft. | Learning Rate (Accuracy/Loss) | | | |
|---|---|---|---|---|
| | 0.01 | 0.001 | 0.0001 | 0.00001 |
| SGD | 0.85/0.8517 | 0.86/0.4871 | 0.84/0.4436 | 0.84/0.4436 |
| RMSprop | 0.79/238.2169 | 0.80/14.8006 | 0.83/1.8811 | 0.85/0.8797 |
| Adagrad | *0.88/0.5842* | *0.88/0.4543* | 0.85/0.4271 | 0.85/*0.4097* |
| Adadelta | 0.86/0.4655 | 0.83/0.4412 | 0.83/0.9728 | 0.79/0.6023 |
| Adam | 0.82/1113.7572 | 0.82/20.2161 | 0.86/0.9728 | 0.86/0.8602 |
| Adamax | 0.82/3.5122 | 0.84/1.2526 | 0.86/0.5242 | 0.86/0.9941 |
| Nadam | 0.82/835.3074 | 0.83/22.9358 | 0.86/0.8873 | 0.84/0.4291 |

**Figure 11.** The Adagrad function (lr = 0.001) confusion matrix using the ISIC 2018 skin lesion datasets.

Table 5 presents the result of the performance comparison between the proposed model and state-of-the-art image classification models. For each of the models compared, the size of the dermoscopic image was normalized to 512 × 512, 256 × 256, and 128 × 128. Meanwhile, for the proposed model, a 1D convolution layer-based model was used with input pattern data of size 50 × 256, generated through an analysis of edge, color, and texture features. The performance of the proposed model was compared with those of DenseNet-201, DenseNet-121, SEResNeXt-101, SEResNeXt-50, ResNet-18, ResNet-50, and ResNet-152. The result shows that the accuracy of the proposed method was 88.6%, which was the highest among all the models compared.

**Table 5.** Performance comparison result with a well-known CNN models for the ISIC 2018 dataset.

| Net | Input Size | Accuracy | Avg. AUC | REF |
|---|---|---|---|---|
| DenseNet-201 | $512 \times 512$ | 86.2% | 98.0% | |
| | $256 \times 256$ | 82.0% | 97.5% | |
| | $128 \times 128$ | 75.2% | 95.0% | |
| DenseNet-121 | $512 \times 512$ | 83.4% | 97.2% | |
| | $256 \times 256$ | 80.9% | 96.6% | |
| | $128 \times 128$ | 73.9% | 94.2% | |
| SEResNeXt-101 | $512 \times 512$ | 85.7% | 98.1% | |
| | $256 \times 256$ | 83.1% | 97.8% | |
| | $128 \times 128$ | 74.5% | 96.0% | |
| SEResNeXt-50 | $512 \times 512$ | 86.7% | 98.2% | |
| | $256 \times 256$ | 81.8% | 97.5% | [10] |
| | $128 \times 128$ | 76.1% | 95.8% | |
| ResNet-18 | $512 \times 512$ | 80.6% | 97.5% | |
| | $256 \times 256$ | 78.9% | 96.8% | |
| | $128 \times 128$ | 70.8% | 93.7% | |
| ResNet-50 | $512 \times 512$ | 84.1% | 97.7% | |
| | $256 \times 256$ | 79.6% | 96.5% | |
| | $128 \times 128$ | 69.2% | 93.0% | |
| ResNet-152 | $512 \times 512$ | 86.1% | 97.8% | |
| | $256 \times 256$ | 80.2% | 97.2% | |
| | $128 \times 128$ | 72.3% | 94.6% | |
| *Our Method* | *$50 \times 256$* | *88.6%* | *94.9%* | - |

In this study, image processing in pixel units was performed to observe skin lesions more closely. The size of the input data generated by this method was $50 \times 256$, which is smaller than that of other models using data with sizes of $512 \times 512$, $256 \times 256$, and $128 \times 128$ pixels. In addition, despite the small input data size, the proposed method contains essential feature information regarding the edge, color, and texture features; therefore, the proposed method enables a higher-level analysis compared with the processing method using only the original dermoscopic image.

Table 6 presents a comparison between methods used in previous studies [33–36] that achieved an excellent performance in skin lesion classification using the ISIC 2018 dataset and the proposed method. As shown, the proposed method outperformed the other methods. Although the result of Minjie shows the highest accuracy at 89.5%, the proposed method is useful and convenient for feature analysis in pixel units, as the method clearly shows the differences of fine-grained features through the detection of edge, color, and texture features, thereby enhancing the classification performance.

**Table 6.** Performance comparison result with a previous related references for the ISIC 2018 dataset.

| REF | Team/Authors | Accuracy | Sensitivity for Melanoma | Avg. AUC | Avg. Specificity |
|-----|--------------|----------|--------------------------|----------|------------------|
| [33] | Nozdryn et al. | 88.5% | 76.0% | 98.3% | 83.3% |
| [34] | Gassert et al. | 85.6% | 80.1% | 98.7% | 98.4% |
| [35] | Amirreza et al. | 86.2% | - | 98.1% | - |
| [36] | Shen et al. | 85.3% | 78.9% | 97.5% | 97.3% |
| [11] | Zhuang et al. | 84.5% | 70.2% | 97.8% | 98.0% |
| | Minjie | 89.5% | 77.8% | 98.2% | 98.1% |
| | Amirreza et al. | 87.4% | 58.5% | 97.9% | 99.2% |
| | MRCN | 87.2% | 86.0% | 95.3% | 97.9% |
| | IPM_HPC | 86.6% | 83.0% | 97.6% | 97.6% |
| | Mahbod | 83.6% | 71.9% | 97.5% | 98.2% |
| | Yao, Peng, et al. (CMEL = 1.0) | 86.4% | 77.2% | 97.4% | 97.1% |
| | Yao, Peng, et al. (CMEL = 1.8) | 85.2% | 86.0% | 97.3% | 96.3% |
| - | *Our Method* | *88.6%* | *71.7%* | *94.9%* | *97.1%* |

Table 7 shows the results of performance validation using the ISIC 2017 dataset. For the existing model [37], a local binary pattern was applied to the original image to obtain the results. As shown in Table 7, the proposed method yielded a high accuracy of 89.3% in melanoma recognition, as well as demonstrating a high performance in terms of the area under the curve (AUC) (87.1%) and sensitivity (SENS) (89.0%). In terms of seborrheic keratosis recognition, the accuracy of the proposed model was low (i.e., 86.3%); similarly, its AUC (91.6%) and specificity (86.3%) were low, although its SENS was higher (86.0%) than that of the other models. Table 8 summarizes the comparison between the results yielded by the proposed method and those reported in previous studies [31,37–39].

**Table 7.** Performance evaluation results of the proposed method and different DCNN models for the ISIC 2017 dataset; ACC (accuracy), SENS (sensitivity), SPEC (specificity).

| Net | Melanoma (%) | | | | Seborrheic Keratosis (%) | | | | REF |
|-----|------|------|------|------|------|------|------|------|-----|
| | ACC | AUC | SENS | SPEC | ACC | AUC | SENS | SPEC | |
| ResNet-50 | 83.0 | 80.4 | 43.6 | 92.5 | 87.7 | 88.0 | 62.2 | 92.2 | [37] |
| DenseNet-121 | 82.8 | 83.5 | 58.1 | 88.8 | 88.3 | 91.0 | 66.7 | 92.2 | |
| ResNet-50 & LBP | 83.8 | 78.4 | 52.1 | 88.6 | 90.5 | 90.9 | 72.2 | 91.3 | |
| DenseNet-121 & LBP | 85.8 | 83.2 | 61.5 | 91.7 | 91.7 | 93.4 | 72.2 | 95.1 | |
| *Our Method* | *89.3* | *87.1* | *89.0* | *89.3* | *86.3* | *91.6* | *86.0* | *86.3* | - |

**Table 8.** Performance comparison result with a previous related references for the ISIC 2017 dataset; ACC (accuracy), SENS (sensitivity), SPEC (specificity).

| REF | Team/Authors | Melanoma (%) | | | | Seborrheic Keratosis (%) | | | |
|-----|--------------|------|------|------|------|------|------|------|------|
| | | ACC | AUC | SENS | SPEC | ACC | AUC | SENS | SPEC |
| [31] | Codella, N.C. et al. | 83.0 | 83.0 | 43.6 | 92.5 | 91.7 | 94.2 | 70.0 | 99.5 |
| [37] | Xiao et al. | - | 85.3 | 67.5 | 90.9 | - | 95.5 | 88.9 | 92.8 |
| [38] | Yang, X. et al. | 83.7 | 85.9 | 59.0 | 89.6 | 90.8 | 95.1 | 77.8 | 93.1 |
| [39] | Zhang, J. et al. | 85.8 | 83.2 | 61.5 | 91.7 | 97.7 | 93.4 | 72.2 | 95.1 |
| - | *Our Method* | *89.3* | *87.1* | *89.0%* | *89.3%* | *86.3%* | *91.6%* | *86.0%* | *86.3%* |

## 5. Conclusions

For the classification of skin lesions in dermoscopic images, we herein proposed a detailed feature extraction method that analyzes edge, color, and texture features in pixel units. The proposed method applies the line-segment-type analysis algorithm to present the line-segment information of edges based on different types. Additionally, the type of visual lines was analyzed in pixel units, and line-segment features were extracted based on cumulative statistics for the same intensity. Regarding color feature extraction for detecting fine-grained changes in color for skin lesions, the image was converted into RGB, HSV, and YCbCr color spaces to analyze the histogram distribution of each channel. Furthermore, to extract texture features, an LTEM filter was used to extract 25 different elements of texture information, and the features were extracted using histogram analysis and a line-segment-type analysis algorithm. Using the input data generated through this process, machine learning was performed via a learning model designed based on a 1D convolution layer. In the designed model, 1D convolution layers with different learning parameters were connected in parallel to determine the optimal weights for the input data.

The performance of the proposed feature data generation method and learning model was compared and evaluated with other well-known classification models using the ISIC 2017 and ISIC 2018 datasets. In experiments using the ISIC 2018 dataset, the proposed method showed a superior performance with an accuracy of 94.9%, compared to previous research results. On the other hand, for the recognition of melanoma and seborrheic keratosis using the ISIC 2017 dataset, the proposed method showed a comparable performance of 97.1% with the other models. Additionally, when compared to results reported in the literature with excellent performance, the proposed method demonstrated an almost equivalent performance.

The method proposed herein can detect fine-grained variations in skin lesion features, and in situations requiring the learning of numerous features, since the proposed method uses extracted data, a reduced computational load can be expected as compared with models using an entire image as the input. However, a significant amount of feature extraction is required for detecting regions of skin lesion with fine-grained features. To effectively extract numerous features, the feature extraction process must be further simplified to increase the speed of data preprocessing and model training. In addition, the experiment revealed that there is a possibility of losing significant feature information during the process of normalizing image sizes, which may result in reduced performance. Therefore, additional experiments are needed to address this issue. The proposed feature extraction method and learning model for detecting and classifying skin lesions may serve as a novel approach for deep learning-based image recognition. Furthermore, they may yield an excellent performance in other fields that require the detection of fine-grained visual differences in input images.

**Author Contributions:** Investigation, M.J., Y.H. (Younghwan Han), Y.H. (Yousik Hong); writing—original draft preparation, C.K.; writing—review and editing, W.L. All authors have read and agreed to the published version of the manuscript.

**Funding:** This research was supported by Sangji University Research Fund, 2020.

**Institutional Review Board Statement:** Not applicable.

**Informed Consent Statement:** Not applicable.

**Data Availability Statement:** Not applicable.

**Conflicts of Interest:** The authors declare no conflict of interest.

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
