# Peer review of "Skin Lesion Classification Using Hybrid Convolutional Neural Network with Edge, Color, and Texture Information"

_applsci, doi:10.3390/app13095497_

Round 1

Reviewer 1 Report

The main scientific achievement of the study is that it shows an increase in the efficiency of object recognition based on feature analysis compared to algorithms using the original images. This is clearly not present in the conclusions, therefore it is better to add it there. Some inaccuracies do not allow the publication of the work in its present form.

Below is a list of comments and suggestions:

  1. There is no clearly stated research goal in the introduction.
  2. References must be performed properly.
  3. References are missing in the text: 24, 33, 34, 35, 36, 37, 38 39.
  4. In line 62: “…Gonzale” – must be “González-Luna”. The whole text should be checked!
  5. In line 124: “…biopsy”, should be reformulated – Most often, visual analysis is sufficient to differentiate pigmented and keratin-like skin lesions. These lesions are completely differentiated using visual dermoscopy. A biopsy is usually not needed! At the same time, a doctor who does not specialize in skin lesions may make a mistake.
  6. In lines 125 – 126: Should be reformulated Dermatofibroma with  a dermoscopy is very easy to differentiate from other skin lesions!
  7. In lines 138-140: Should be reformulated. A biopsy is done for differential diagnosis in difficult cases. For melanocytic lesions with suspected melanoma, excision biopsy is characteristic, due to the tendency of such lesions to malignate, and melanomas to metastasize. If BCC or SCC is suspected, a biopsy is rather needed to clarify the subtype of cells in the formation of subsequent treatment tactics. In the absence of suspicion of oncology, a biopsy is not performed.
  8. In lines 143-145: Should be reformulated.  Dermoscopic signs are known in medicine, which make it possible to unambiguously differentiate classes of lesions (pigmented, non-pigmented, vascular, etc.), as well as to differentiate within classes. The accuracy of visual dermoscopy is at least 95%. Visual dermoscopy is today the gold standard for the preliminary diagnosis of skin lesions.
  9. Figure 1: check the correspondence of pictures and captions – there are inconsistencies.
  10. In line 180: “…authors in a previous study” – Should be referenced.
  11.  In lines 189 – 190: Should be reformulated. With a dermoscopy a specialist can accurately assess the signs of lesions and make a clinical diagnosis with an accuracy above 95%.
  12. In lines 196-197: Should be reformulated.  In clinical practice, internal and external borders are not different.
  13. The border on dermoscopic images can bee not closed, because the formation may exceed the field of view.
  14. In lines 250-252: Should be reformulated. The lesion begins where the skin ceases to be normal. The internal border characterizes the internal features of the lesion, and the area of the lesion between the internal and external borders is actually the border of the lesion – the transition from normal skin to the tissues of the lesion.
  15. Section 3.1. The described threshold processing method for detecting lesion border is incorrect, since it will not give the true lesion border visible to the eye. Such algorithms should be compared with expert border detection. The described method rather evaluates some "extreme" features of the texture in the area of the lesion border, although it does not detect the border  themselves (in the accepted modern interpretation). At the same time, as a method of feature formation for machine learning, this method is applicable.
  16. In line 272: It can be clarified that the reduction is obtained when the pattern are grouped when turning.
  17. Error in equation 4: Sh=[M/a], must bee N.
  18. In line 290: why 16 regions?
  19. In line 292: EFM  must bee EFA.
  20. Figure 7: Images for different color systems should look the same when presented on any medium, since this is only a method to encode color information. The figure maybe shows some color channels for HSV and YCbCr?
  21. Equation 7: Mixj – x should be removed.
  22. In line 372: 256 × 256 and later – Converting source images to a reduced resolution 256 × 256 (128 × 128, 512 × 512) leads to the loss of important diagnostic information, because important textural features of dermoscopic images are lost. This is probably why the proposed approach and other approaches in the article demonstrated such a low classification accuracy: less than 90%, which is lower than with visual dermoscopy (95%). This is worth adding in conclusion.

Processing comments can significantly improve the article. Good luck!

Author Response

Dear reviewer.

Please find our revised manuscript entitled “Skin lesion classification using hybrid convolutional neural network with edge, color, and texture information” submitted for consideration for publication in Applied Sciences.

Thanks for the reviewers through review of our manuscript, which greatly improved the quality of the manuscript.

On the separate sheets, the reviewer's very helpful comments were addressed, point by point, in our response to each comment or revision request, with an indication of pages on which the manuscript changes have been made.

We hope that after these corrections our manuscript will be acceptable for publication in Applied Sciences.

Sincerely yours.

Reviewer 2 Report

The author addressed the hybrid convolutional neural network for skin lesion classification with edge, color and texture information. 

Based on my review, I have some questions/suggestions to improve the quality of the paper. 

1.  I suggest the author highlight the contribution in the introduction section. 

2. Include the structure of the paper at the end of the introduction section. 

3. English and grammatical check is required. 

Author Response

(The authors gave the same response as above.)

Reviewer 3 Report

Dear authors

Congratulations on the article. Here are some simple suggestions for a better presentation of the study:

1) Line 406 – It is suggested that the experimental results presented in Table 4 inform which was the loss function used. As well as informing whether the “random state” of the machine was constant or random for each simulation performed.

2) Line 408 - What are the criteria for using the 80% and 20% percentages, for training and testing, respectively?

3) Line 454 - Based on the results presented, it is necessary to improve the Conclusion.

Best Regards

Author Response

(The authors gave the same response as above.)
